# Characterisation of Extracts Obtained from Unripe Grapes and Evaluation of Their Potential Protective Effects against Oxidation of Wine Colour in Comparison with Different Oenological Products

**DOI:** 10.3390/foods10071499

**Published:** 2021-06-28

**Authors:** Giovanna Fia, Ginevra Bucalossi, Bruno Zanoni

**Affiliations:** DAGRI—Department of Agricultural, Food, Environmental, and Forestry Sciences and Technologies, University of Florence, Via Donizetti, 6-50144 Firenze, Italy; ginevra.bucalossi@unifi.it (G.B.); bruno.zanoni@unifi.it (B.Z.)

**Keywords:** antioxidant extracts, unripe grapes, wine colour, anti-browning effect, phenols, SO_2_, ascorbic acid, oenological tannins

## Abstract

Unripe grapes (UGs) are a waste product of vine cultivation rich in natural antioxidants. These antioxidants could be used in winemaking as alternatives to SO_2_. Three extracts were obtained by maceration from Viognier, Merlot and Sangiovese UGs. The composition and antioxidant activity of the UG extracts were studied in model solutions at different pH levels. The capacity of the UG extracts to protect wine colour was evaluated in accelerated oxidation tests and small-scale trials on both red and white wines during ageing in comparison with sulphur dioxide, ascorbic acid and commercial tannins. The Viognier and Merlot extracts were rich in phenolic acids while the Sangiovese extract was rich in flavonoids. The antioxidant activity of the extracts and commercial tannins was influenced by the pH. In the oxidation tests, the extracts and commercial products showed different wine colour protection capacities in function of the type of wine. During ageing, the white wine with the added Viognier UG extract showed the lowest level of colour oxidation. The colour of the red wine with the UG extract evolved similarly to wine with SO_2_ and commercial tannins. The obtained results indicated that natural and healthy UG extracts could be an interesting substitute for SO_2_ during wine ageing.

## 1. Introduction

Sulphur dioxide (SO_2_) is a chemical additive widely and traditionally used to preserve wine from oxidation and microbiological spoilage. SO_2_ is very efficient at preventing chemical and enzymatic oxidation and microbial alterations, as it is effective against bacteria, yeasts and moulds naturally occurring in grapes and wine, and against sensory quality deterioration, helping to maintain the colour and aroma of the product. However, SO_2_ in food and beverages is associated with several human health risks including dermatitis, urticaria, abdominal pain, bronchoconstriction and anaphylaxis [1]. Other preservatives, such as ascorbic acid and potassium sorbate, can be used in association with SO_2_ to preserve wine from oxidation or to prevent fermentation of wines with residual sugars. An exhaustive list of additives used in winemaking is reported on the International Organization of Vine and Wine (OIV) website [2]. Consumers’ increasing demands for “natural” and “healthy” wines has induced winemakers to lower the amount of SO_2_ in wine or to find alternative solutions to sulphur dioxide [3]. Moreover, in 2011 the European Community established that a SO_2_ content of more than 10 mg/L should be declared on the wine label. 

Oenological tannins are commercial plant extracts of different botanical origins, for example, from oak galls, chestnut, quebracho, acacia, grape seeds and skins, tara and mimosa. They represent an interesting alternative to SO_2_ for the protection of wine against oxidation. Oenological tannins inhibit polyphenol oxidases, tyrosinase (EC 1.14.18.1) and laccase (EC 1.10.3.2), thus protecting musts and wines from browning. They also directly consume oxygen, thereby protecting the other wine components from oxidation [4,5,6]. 

Recently, new extracts of plant origin such as black radish, almond skin, eucalyptus leaves, grapevine shoots and grape stems have been proposed as alternatives to SO_2_, with interesting results [7,8,9,10,11,12,13]. However, most of these studies were conducted in model solutions or in wine at lab scale, while very few were carried out during winemaking at the winery scale. Moreover, some of these innovative antioxidants affect the wine composition and sensory quality [12]. Therefore, despite the promising results, the use of these alternative antioxidant products is not yet widespread in the wine industry. 

Nowadays, the wine industry is faced with another challenge concerning the huge amount of waste generated by the supply chain, mainly consisting of industrial residue (pomace and lees) and residue from vine cultivation such as stalks, leaves and vine cane [14,15]. The reuse of these waste materials to obtain new products or energy is one of the fundamental actions in the race to achieve sustainable production systems. Unripe grapes (UGs) are a waste product deriving from thinning, which is a green pruning practice carried out on the vines to improve the composition of the grapes to produce high-quality red and white wine [16,17]. The UGs removed from the vine are normally abandoned in field and left to rot. Recently, extracts from UGs were tested for different applications as potential antioxidants, acidifying and antibacterial ingredients in food and beverages [18,19,20,21,22,23,24,25]. Grapes are rich in antioxidant compounds such as phenols, stilbenes, glutathione and vitamins and the concentration of antioxidant compounds varies according to several factors such as the variety, vintage, climatic condition and ripening stage [26,27,28,29,30]. There are no specific rules to decide the optimal timing for cluster thinning [31], but around *veraison* seems to be the most effective moment. *Veraison* is a crucial stage in the grape’s evolution which approximately coincides with the start of the normal expansion in berry volume, accumulation of sugars and anthocyanins in red grapes, and decrease in acidity. During the herbaceous growth phase that precedes *veraison*, the berries are small, green and hard [32] and their acid content increases due to the accumulation of tartaric and malic acids in the pulp cells. Malic acid accumulates rapidly, reaching high values of up to 25 g/L of juice, while the accumulation of tartaric acid is slower, although its concentration can be as high as 15 g/L at the end of the herbaceous growth phase [33,34,35,36]. After *veraison*, at the onset of ripening, the acidity begins to decline, mainly due to the consumption of malic acid as a substrate for the respiratory metabolism. Organic acids can exert an important inhibitory effect against polyphenol oxidases (PPO), which are a group of copper-containing enzymes, found widely in animals, plants, fungi and bacteria, and responsible for the browning of damaged fruit and vegetables, with detrimental effects on the quality of these products [37].

Grapes show a very complex phenol compound profile. Most of them are very efficient radical scavengers, contributing to the antioxidant power of juice and wine. Flavonoids are the most abundant phenol compounds in grapes. The different classes of flavonoids (anthocyanins, flavonols and tannins) are mainly located in the grape skin and seeds and display different evolution patterns during ripening [26,38,39,40]. Anthocyanin accumulation in the skin of red grapes begins at *veraison* and continues during ripening. Flavonol biosynthesis occurs in two distinct periods, first around flowering and second after *veraison* and continues in the berry skin throughout ripening [40]. Tannins are found in both the berry skin and seeds. In the seeds, tannin accumulation starts immediately after fruit set and reaches a maximum level around *veraison*, while in the skins tannin accumulation continues from fruit set until one or two weeks after *veraison* [26,41,42,43,44,45,46,47,48]. The phenolic acids of grapes are less concentrated compared to the flavonoids and they are mainly located in the cells of the pulp [26,39]. Hydroxycinnamic acids and their esters with tartaric acid are the most abundant phenolic acids in grapes and, on a per-berry basis, they show peak concentration prior to *veraison* followed by a decline during ripening [40]. The cinnamoyl esterases, a subclass of carboxylic ester hydrolases (EC 3.1.1.1), are responsible of the hydrolysis of hydroxycinnamic tartaric esters and the consequent release of the hydroxycinnamic acids [49]. The antioxidant compounds found in grapes and intensively studied for their positive health effects [50] are resveratrol and its oligomer forms (viniferins) [51]. These stilbenes are an important group of phenolic substances that exist naturally in different tissues of the grapevine, such as the berry (skin and seeds), leaves and canes. The amount of resveratrol is high in UGs while it drops to low levels in the ripe fruit [26,28].

Other potent antioxidant compounds can be found in grapes. One of these is glutathione (GSH), a tripeptide that begins to accumulate at the onset of ripening. GSH concentration in the grapes is correlated with sugar accumulation during ripening. At maturity, the GSH concentration ranges from 0 to 100 mg/L according to several factors, such as grape cultivar and environmental conditions. When the grape juice is exposed to oxygen, GSH is rapidly oxidised. Two forms of glutathione can be found in grape juice, glutathione disulphide (GSSG) deriving from the enzymatic oxidation of GSH and 2-S-glutathionil caftaric acid (grape reaction product-GRP) which is formed when the grape juice is exposed to oxygen and the *o*-quinones deriving from the enzymatic oxidation of phenolic acids are in the presence of GSH [52]. 

Grapes contain liposoluble and water-soluble vitamins. Water-soluble vitamins generally display polar or ionizable groups, whereas liposoluble vitamins are more commonly characterised by aromatic and aliphatic groups [53]. Liposoluble vitamins include vitamin E and carotenoids. Vitamin E consists of four tocopherols (α-, β-, γ- and δ-forms) and the corresponding unsaturated forms, tocotrienols. Tocopherols are concentrated in grape seeds and the alpha form is predominant [54]. Tocopherols decrease while tocotrienols accumulate during seed development [55]. As they are powerful antioxidants, tocopherols prevent oxidation of the fats and oil contained in the seeds, ensuring their longevity and healthy germination [56]. The water-soluble vitamins in grapes include vitamin C, pantothenic acid, niacin, riboflavin, thiamine, biotin and choline [57]. 

With the aim of finding a healthy and sustainable alternative to sulphur dioxide, we characterised the composition of three UG extracts and their antioxidant activity in a model solution. Moreover, we tested the capacity of UG extracts to protect wine colour in both white and red wines in the laboratory using accelerated oxidation tests [58], and compared them with sulphur dioxide, ascorbic acid and commercial tannins. Then, we tested the UG extracts as colour protecting additives for red and white wines in trials performed on small volumes during ageing in stainless steel tanks. 

## 2. Materials and Methods

### 2.1. Chemicals

All reagents and solvents were purchased from Sigma-Aldrich (Milan, Italy), except for methanol and ethanol which were supplied by Carlo Erba (Milan, Italy) and quercetin-3-*O*-glucoside, quercetin-3-*O*-glucuronide and rutin, which were supplied by Analytik GmbH (Rülzheim, Germany). Ultrapure water was obtained using a Milli-Q Gradient water purification system (Thermo Scientific, Waltham, Massachusetts, United States). Potassium metabisulphite and citric acid were purchased from Carlo Erba (Milan, Italy). Commercial oenological tannins extracted from oak (*Quercus alba*) were purchased on the market.

### 2.2. Unripe Grape Extracts

The extracts were obtained from UGs of three different varieties (Viognier, Merlot and Sangiovese) following the method described by Fia et al. [21]. Briefly, the unripe grapes were handpicked, destemmed and crushed in the wine cellar. The crushed grapes were extracted by maceration for 96 h at 6 °C with the addition of dry ice. After racking, the liquid extract was decanted for 48 h at 6 °C and filtered in order to remove any large particles (i.e., 1 mm or more in diameter). In the case of the Viognier and Merlot extracts, the sugars were eliminated by ultrafiltration, using a spiral wound configuration membrane, with a molecular weight cut-off of 2500 Da (General Electric, Boston, MA, USA). The liquid extracts from the Merlot and Viognier UGs were dehydrated by lyophilization with the addition of Arabic gum (2% *w*/*v*) (Nexira Food, Rouen Cedex, France) while the liquid extract from the Sangiovese UGs was spray-dried with the addition of Arabic gum (16% *w*/*v*). The dried extracts were stored under vacuum in polyethylene pouches, at room temperature and protected from the light.

### 2.3. Wines

Young commercial wines—11 red and 6 white—were purchased from large-scale distribution channels or supplied by producers. The 11 red wines belonged to three different types: conventional (5), organic (3) and biodynamic (3). The 6 white wines belonged to two types: conventional (3) and organic (3). Eleven samples were bottled, four samples were conditioned in Tetra Brik and two samples in a bag-in-box. 

### 2.4. Model Solution

The model buffer solution was prepared by dissolving 5 g/L of tartaric acid in distilled water—ethanol mixture (12% *v*/*v*) and the pH was adjusted to 3.2 and 3.5 by adding sodium hydroxide. A 500 mg/L dose of sulphur dioxide, ascorbic acid, commercial tannins and UG extracts was added to the model buffer solution and the antioxidant activity of the samples was immediately evaluated. 

### 2.5. Small-Scale Trials

We performed small-scale trials using red and white wines. The red wine was taken at the end of malolactic fermentation. Both wines were chosen based on their sulphur dioxide content, which was not detectable in the red wine and below 10 mg/L of free SO_2_ in the white wine. We used a 300 L volume of both the red and white wines. Each trial was arranged in duplicate, by transferring aliquots (50 L each) into six stainless steel tanks for each wine. The SO_2_ was added to the wine at a ratio of 100 mg/L, while commercial tannins and UG extracts were added at a ratio of 400 mg/L. Viognier UG extract was used for the white wine, while Sangiovese UG extract was added to the red wine. All the wines were maintained at a temperature of 18 °C, for 90 days. Two decantations were performed after 30 and 60 days. The evolution of the wine colour was followed by spectrophotometric analysis which measured the absorbance at 420, 520 and 620 nm.

### 2.6. General Analysis

The total acidity, volatile acidity, pH, alcohol, and total and free SO_2_ were evaluated according to the methods recommended by the International Organization of Vine and Wine (OIV) [59]. 

### 2.7. Total Phenols

The total phenols (TPs) were quantified according to the Folin–Ciocalteu method [60], with modifications. Before performing the Folin–Ciocalteu assay, the phenolic compounds were removed from 1 mL of undiluted wine or 10% extract solution with a C18 Sep-pak cartridge (Waters, Milan, Italy), following the method described by Di Stefano et al. [60]. Four mL of sodium carbonate (10%, *w*/*v*) were added to 1 mL of each sample, mixed well, and left to stand for 5 min. A volume of 1 mL of diluted FC reagent was added to the mixture and the samples were left in the dark for 90 min at room temperature. Then the absorbance was measured spectrophotometrically at 700 nm with a Perkin Elmer Lambda 10 spectrophotometer (Waltham, MA, USA). A standard curve was obtained with (+)-catechin solutions at concentrations ranging from 5 to 500 mg/L. The TP content was expressed as mg of (+)-catechin equivalents (mg CAT eq)/g for the UG extracts and (mg CAT eq)/L for the wine.

### 2.8. Total Polyphenol Index

The Total Polyphenol Index (TPI) was determined by measuring the absorbance at 280 nm of an aqueous dilution of red wine in a 10 mm quartz cuvette using a Perkin Elmer Lambda 10 spectrophotometer (Waltham, MA, USA). The absorbance value was then multiplied by the dilution factor [61].

### 2.9. Colour Intensity and Hue

The colour intensity (CI) and hue (H) of the undiluted wine were measured using a 1 mm path-length quartz cell with distilled water as a reference [62]. The absorbance at 420, 520 and 620 nm was measured using a Perkin Elmer Lambda 10 spectrophotometer (Waltham, MA, USA). For the red wine, the CI was calculated using the following formula:CI = (A420 +A520 + A620) × 10

H was calculated as follows:H = A420/A520

### 2.10. Phenolic and Glutathione Content

The phenolic composition and glutathione content were measured with liquid chromatography high-resolution mass spectrometry (LC-HRMS) following the method described by Fia et al. [21]. A chromatograph Accela 1250 (Thermo Fisher Scientific, Waltham, MA, USA), a Kinetex F5 column (2.1 × 100 mm 1.7 μm-Phenomenex (Torrance, CA, USA)) and an LTQ OrbitrapExactive mass spectrometer (Thermo Fisher Scientific, Waltham, MA, USA) were used. The quantitative analysis was performed with TraceFinder^TM^ 4.1 software (Thermo Fisher Scientific, Waltham, MA, USA) following an external standard method, using a linear regression from 0.05 to 1 g/L of five standard solutions.

### 2.11. Antioxidant Activity

The antioxidant activity (AA) was determined by a 2,2-diphenyl-1-picrylhydrazyl (DPPH) spectrophotometric assay [63]. In brief, a solution of DPPH (6 × 10^−5^ M) was prepared by dissolving 0.236 mg of DPPH in 100 mL of methanol. For the reaction, 0.1 mL of the sample was mixed with 3.9 mL of DPPH stock solution. For the reference sample, 0.1 mL of the sample was added to 3.9 mL of methanol. The maximum absorbance of DPPH was tested by mixing 0.1 mL of methanol with 3.9 mL of DPPH stock solution. The samples were left in the dark for 30 min at 30 °C for the reaction. Immediately afterwards, the decrease in absorbance at 515 nm was measured with a Perkin Elmer Lambda 10 UV-Vis spectrophotometer (Waltham, MA, USA), against the methanol reference sample. Trolox standard solutions were prepared daily in absolute ethanol at concentrations ranging from 10 to 600 μmol/L. The antioxidant activity was expressed as mmol of the Trolox equivalent antioxidant capacity (mmol TEAC/L) while specific antioxidant activity was calculated and expressed as mmol TEAC /g of phenols or active substances (ASs), where these were sulphur dioxide and ascorbic acid.

### 2.12. Accelerated Oxidation Tests

The predisposition towards browning was determined by the polyphenols oxidative medium (POM) test for the white wine and the anthocyanin oxidability index (AOI) test for the red wine [58,64,65]. Increasing doses of a 3% hydrogen peroxide solution were added to 15 mL of wine. For the white wines, the hydrogen peroxide doses ranged from 62.5 to 4000 μL/L, while they ranged from 500 to 8000 μL/L for the red wines. After 1 h at 60 °C, the percentage of colour oxidation (OX%) produced was estimated. The percentage increase in absorbance at 420 nm was considered in the case of white wine while the percentage decrease in absorbance at 520 nm was considered for the red wine. The absorbance was measured using a Perkin Elmer Lambda 10 spectrophotometer (Waltham, MA, USA) before oxidation (A_BOX_) and after oxidation (A_AOX_).

The percentage increase in absorbance at 420 nm in the white wines was calculated using the following formula:OX% = [(A_420AOX_−A_420BOX_)/A_420BOX_] × 100

In the case of red wine, the percentage decrease in absorbance at 520 nm was calculated using the following formula:OX% = [(A_520BOX_−A_520AOX_)/A_520BOX_] × 100

### 2.13. Statistical Analysis

Statistical analysis was carried out using the XLSTAT statistical and data analysis solution (2021) (Addinsoft, Paris, France). A one-way ANOVA (Least Significant Differences (LSD), 5% level) was performed on the phenol composition and antioxidant activity to check significant differences among the UG extracts. Principal component analysis (PCA) was performed on the wine parameters to extract the most relevant information. Pearson’s correlation coefficient was carried out to determine correlations between the antioxidant assay the other chemical and physical parameters of the wines. 

## 3. Results

### 3.1. Unripe Grape Extract Characterisation

#### 3.1.1. Composition of the UG Extracts

The composition of the UG extracts is shown in Table 1. Twenty-five compounds, including 22 phenols, grape reaction product (GRP), glutathione and reduced glutathione (GSSG) were identified by LC-HRMS. The total phenol (Σ*_Phenols_*) content of the extracts was calculated as the sum of the amount of each phenol compound. The total phenols in the Viognier UGs (1691 μg/g) were higher than in the Merlot (1336 μg/g) and Sangiovese (1045 μg/g) extracts. The phenolic composition of the UG extracts was similar while the concentration of the different compounds and the total amount of the different phenol classes varied according to the type of extract. The Sangiovese extract had the highest concentration of flavan-3-ols, procyanidins and flavonols, while the Merlot and Viognier extracts had the highest concentration of phenolic acids. Phenolic acids represented about 90% and 96% of the total phenols in the Viognier and Merlot extracts, respectively (Table 1). In the Sangiovese extract, phenolic acids only represented about 50% of the total phenols. The Sangiovese extract was significantly richer in (+)-catechin, (−)-epicatechin, procyanidin B1 and B2, quercetin-3-*O*-glucuronide and quercetin-3-*O*-hexoside with respect to the Merlot and Viognier extracts. In the Sangiovese extract, (+)-catechin was more than 15 times more concentrated than in the Merlot and about 3.5 times more than the Viognier extract. 

Caftaric acid, ferulic acid and glutathione were significantly higher in the Merlot and Viognier than in the Sangiovese extract. Caftaric acid alone represented about 70% and 80% of the total phenols in the Viognier and Merlot extracts. Viognier had the highest content of fertaric acid and GRP while Merlot had the highest resveratrol content. Myricetin was only detected in the Merlot extract. Glutathione was detected in the Viognier and Merlot extracts while GSSG was only detected in the Sangiovese extract. 

The total phenol (TP) content of the UG extracts was measured by Folin–Ciocalteu assay. The total phenol content was 20.4 ± 0.2 (Merlot extract), 10.4 ± 0.2 (Viognier extract) and 8.7 ± 0.6 (Sangiovese extract) mg CATeq/g of powder.

#### 3.1.2. Antioxidant Activity of the UG Extracts and Oenological Products

The antioxidant activity of the UG extracts, sulphur dioxide, ascorbic acid and commercial tannins was tested preliminarily in a model buffer solution at pH 3.2 and 3.5 and expressed as specific antioxidant activity (AA) (Figure 1). The two pH values were chosen to simulate white (pH 3.2) and red (pH 3.5) wine. The AA of the model buffer solution was very low and can be considered negligible. In contrast, the specific AA was influenced by the type of extract, commercial product and pH. 

At pH 3.2, the Sangiovese extract gave a significantly higher value of specific AA, followed by the Merlot extract, ascorbic acid, the Viognier extract, commercial tannins and SO_2_. At pH 3.5, a different ranking was observed. Sangiovese had a significantly higher value of specific AA followed by the Merlot extract, commercial tannins, the Viognier extract, ascorbic acid and SO_2_. At both pH values, the specific AA of the Sangiovese extract was about 10 times higher than that of the Merlot extract and almost 30 times higher than that of the Viognier extract. Moreover, at pH 3.5, all the UG extracts and commercial tannins showed similar behaviour, with higher specific AA values with respect to the levels observed at pH 3.2 while the specific AA of ascorbic acid and SO_2_ did not seem to be influenced by the variation in pH.

To compare the effectiveness of the commercial products and UG extracts as radical scavengers, the specific AA of each product was referred to the doses of products usually added to wine. To this end, we considered 100 mg/L for SO_2_, 100 mg/L for ascorbic acid and 400 mg/L for the tannins and UG extracts (Table 2). Then, the percentage relative antioxidant activity (RAA) of ascorbic acid, commercial tannin and the UG extracts with respect to SO_2_ was calculated by setting the specific AA of SO_2_ as 100%. All the commercial products and UG extracts had a higher percentage RAA compared to SO_2_ at both pH values. The Sangiovese and Merlot extracts were most effective at both pH levels, followed by tannins at pH 3.5, the Viognier extract at pH 3.5 and 3.2, commercial tannins at pH 3.2 and ascorbic acid at both pH values.

### 3.2. Composition of the Wines

To test the different ability to counteract oxidation through the addition of hydrogen peroxide, we used white and red wine samples with very different compositions in terms of total and free SO_2_, pH, total acidity (TA), volatile acidity (VA), alcohol, total phenol (TP), total phenol index (TPI), Abs 420, 520 and 620, colour intensity (CI), colour hue (H) and antioxidant activity (AA).

The principal component analysis (PCA) of the chemical parameters of the white wines is shown in Figure 2. The two principal components (F1 and F2) account for 78% of the variability among the samples. The conventional wines (1W, 2W and 3W) conditioned in Tetra Brik or a bag-in-box are grouped on the left side of the graph while the conventional bottle-conditioned sample (6W) is separate. The organic samples (4W and 5W) are on the opposite side of the graph to the conventional ones. The chemical parameters of the 6 white wines varied over a wide range: pH (3.02–3.58), total acidity (2.7–6.3 g/L tartaric acid equation), volatile acidity (0.30–0.76 g/L acetic acid equation), alcohol content (10.6–13.6% *v*/*v*), free SO_2_ (13–77 mg/L), total SO_2_ (683–149 mg/L), total phenols (0.15–1.40 g/L), antioxidant activity (0.47–15.0 mmol TEAC/L) and Abs 420 (0.04–0.30 absorbance units). The content of total and free SO_2_, which was higher in 1W, 2W and 3W compared to the other samples, did not seem to be correlated with the antioxidant activity of the wines. On the contrary, the total phenol content and total acidity of the wine samples were positively correlated with the radical scavenging activity of these white wines (Appendix A).

The principal component analysis of the chemical parameters of the red wines is shown in Figure 3. The two principal components (F1 and F2) accounted for about 65% of the variability among the samples. Samples 1R, 2R and 3R, more or less grouped on the left of the graph, were conventional wines conditioned in Tetra Brik or a bag-in-box while the other conventional bottle-conditioned wines (4R and 5R) are separate. The 6R, 7R and 8R wines were organic, while 9R, 10R and 11R were biodynamic samples. The chemical parameters of the 11 red wines varied over a wide range: pH (3.14–3.96), total acidity (4.0–6.3 g/L tartaric acid equation), volatile acidity (0.28–0.92 g/L acetic acid equation), alcohol content (11.5–14.3% *v*/*v*), free SO_2_ (0–49 mg/L), total SO_2_ (14–104 mg/L), total phenols (2.11–5.57 g/L), antioxidant activity (0.35–23.1 mmol TEAC/L) and colour intensity (3.8–8.6 absorbance units). Similar to what was observed for the white wines, the radical scavenging activity of the red wines was positively correlated with the total phenol content rather than with the SO_2_ content (Appendix A). Sample 8, an organic wine elaborated without the addition of SO_2_, showed the highest AA.

### 3.3. Accelerated Oxidation Tests

The POM test was performed by adding 8 different doses of 3% H_2_O_2_, from 62.5 to 4000 μL/L of wine to the white wines. The red wines were treated with 10 different doses of 3% H_2_O_2_, ranging from 500 to 16,000 μL/L of wine. The results of these tests are reported in Figure 4 and Figure 5.

After 1 h of incubation at 60 °C, the wines clearly showed a different resistance to colour oxidation as a consequence of the addition of hydrogen peroxide. For example, the OX% obtained after the addition of 4000 μL/L of H_2_O_2_ (dose usually used in these tests) [58,64,65] varied from 53 to 100% in the white wines and from 7 to 69% in the red wines. Moreover, it is possible to observe that different wines reached different maximum levels of colour oxidation, after which further additions of H_2_O_2_ did not lead to a further increase in the oxidation percentage and the curve tended to plateau. 

For the purpose of comparing the results of the POM and AOI tests on the wines with added UG extracts and commercial products, first, we standardized the amount of hydrogen peroxide necessary to give a similar level of oxidation to all the wine samples to which nothing had been added. For the white wines, we chose the amount of H_2_O_2_ able to produce an oxidation percentage of 40 to 60% while for the red wine we considered an oxidation percentage of 30 to 50%. Commercial products were added at a concentration of 100 mg/L (SO_2_), 100 mg/L (ascorbic acid) and 400 mg/L (tannins), while for the UG extracts it was necessary to achieve a concentration of 2 g/L to obtain lower levels of OX% than that of the wine to which nothing had been added. The results of the POM tests performed by adding the chosen amount of H_2_O_2_ to white wines containing commercial products and UG extracts are shown in Figure 6.

Ascorbic acid, followed by SO_2_, had the highest capacity to protect the colour of the white wines. Commercial tannins showed the worst performance compared to the other products in the POM tests. All the UG extracts protected the colour of the white wines against oxidation by hydrogen peroxide. Sangiovese gave the lowest oxidation percentage value, followed by the Merlot and Viognier UG extracts. 

The results of the AOI test performed by adding the chosen amount of H_2_O_2_ to red wines containing commercial products and UG extracts are shown in Figure 7. In the red wines, the antioxidant products showed different behaviour with respect to what was observed in the white wines. Indeed, SO_2_ had the highest antioxidant capacity while ascorbic acid did not protect the colour of the red wine but seemed to stimulate its oxidation. The UG extracts and commercial tannins had a similar antioxidant capacity, which was lower than that of sulphur dioxide but higher than that of ascorbic acid. In the majority of the samples, the antioxidant capacities of the UG extracts were similar.

### 3.4. Small-Scale Trials

In the small-scale trials, we used the doses of antioxidant products usually added to wine. The SO_2_ was added to the wine at a ratio of 100 mg/L while 400 mg/L doses of commercial tannins and UG extracts were added. The Viognier UG extract was chosen for the white wine, mainly because of its clear colour which consequentially had less impact on the colour of the wine (Figure 8 and Figure 9) and because this extract did not contain sugar.

The evolution of absorbance at 420 nm of the white wine sample is shown in Figure 10. Immediately after the product additions, the sample with commercial tannins had the highest A420 value followed by the wines with the Viognier UG extract and SO_2._ At 30 days, immediately after the first decantation, the A420 value was similar to the initial measurement, while after 60 days, at the second decantation, the A420 values started to increase. The effect of the second decantation was evident after 90 days when the A420 value of all the samples was higher than previous measurements. At this time, the sample containing the Viognier UG extract had the lowest value of A420 and the percentage increase in A420 in the wine with SO_2_ was approximately twice that measured in the other samples.

The Sangiovese UG extract was chosen for the small-scale trials on the red wines, mainly because this extract showed the highest specific AA in the preliminary tests at pH 3.5. Immediately after the product additions, the wines with commercial tannins and Sangiovese UG extract had the highest A520 value while the wine with SO_2_ had the lowest (Figure 11). After 30 days, immediately after the first decantation, the A520 values of all the wines were similar. A reduction in the A520 was observed in all the samples at 60 days, immediately after the second decantation while, in the end, the percentage variation of A520 was similar to that observed after 30 days. At the end of the tests, there were no significant differences among the samples. The evolution of the colour intensity and hue of the red wines over 90 days is shown in Figure 12. Immediately after addition, the sample with SO_2_ showed the lowest CI and H. After that, the colour of the red wine evolved similarly in all the tested conditions and, after 90 days, there were no differences among the samples. Therefore, all the products (SO_2_, commercial tannins and Sangiovese UG extract) gave the red wine the same protection against colour oxidation.

## 4. Discussion

The phenolic substances of grapes include flavonoids (anthocyanins, flavonols and tannins), phenolic acids and stilbenes. Phenol compounds of grapes belong to many different chemical structures, from simple and low molecular weight compounds such as those of phenolic acids, to very complex and high molecular weight compounds such as those of tannins that derive from the polymerization of flavan-3-ols [26,28,66,67,68]. The proportion of the different classes of phenols changes as the grapes ripen, from fruit set to commercial maturity. The phenolic composition of grapes is influenced by the variety [26,27,28]. The phenolic composition of the UG extracts was similar to that observed by other authors who have analysed juices and extracts obtained from unripe grapes. The phenol profile of seven juice samples highlighted the complex composition of unripe grapes, containing hydroxycinnamic and benzoic acids, flavan-3-ols and flavonols [69]. The juices obtained by pressing unripe grapes (cv. Merlot and Barbera) from two vintages contained flavan-3-ols, mainly epigallocatechin gallate, and phenolic acids [25]. Phenolic acids, flavonols, flavan-3-ols, procyanidins (B1 and B2) and resveratrol were detected in the liquid extracts obtained from Sangiovese unripe grapes by Fia et al. [21] and Fia et al. [57]. In the Sangiovese extracts, caftaric and fertaric acid, (+)-catechin, (−)-epicatechin and quercetin-3-*O*-glucunoride were the most abundant phenol compounds [57]. The differences highlighted in the concentration of the different phenolic compounds of the three UG extracts could be due to the variety of grapes. It is known that the Sangiovese grape variety is particularly rich in quercetin and its derivatives [70,71]. However, the phenolic composition of the extracts may also have been influenced by the drying method. The freeze-drying method seems to preserve the phenol compounds better than the spray-drying method [72,73]. 

The phenol content of the UG extracts assayed using the Folin–Ciocalteu method was on average about 10 times higher than the sum of the single phenol compounds evaluated by LC-HRMS. Similar results were obtained by another author who measured the total phenols in unripe grape juices by Folin–Ciocalteu and calculated the total phenol content as a sum of the polyphenols assayed by High Performance Liquid Chromatography (HPLC) [69]. Since this author noticed large discrepancies between the total polyphenol values obtained using the two different methods, he hypothesized that most of the phenolic fraction of unripe grapes could be of a polymeric nature. Some evidence supported the fact that green grapes can contain a high amount of tannins localized in the berry seeds and skins [26,30].

In the model solution at different pH values, the higher specific AA shown by the Sangiovese compared to that of the Merlot and Viognier UG extracts could be due to the different composition in terms of phenolic compounds. In particular, (+)-catechin (−)-epicatechin, procyanidin B1 and B2 and flavonols, which were more concentrated in the Sangiovese extract, could have contributed to the observed differences in the specific antioxidant activity [74]. The radical scavenging activity of phenol compounds is influenced by their structure and, with some exceptions, phenolic acids are less effective than flavonoids [75,76,77]. The number and position of hydroxyl groups and chemical substituents linked to aromatic rings are important chemical characteristics that influence the antioxidant capacity of flavonoids. For example, among the different grape flavonols, myricetin has a lower antioxidant activity compared to quercetin but higher compared to kaempferol; these compounds differ in the substitution pattern on the B ring [78,79,80,81]. Furthermore, the antioxidant capacity of flavonoids increases as their degree of polymerization increases, for example, proanthocyanidins, the polymers of catechins and condensed tannins, are excellent in vitro antioxidants due to the high number of hydroxyl groups in their molecules [82]. 

The Sangiovese UG extract and commercial tannins in the model solution were affected by the pH in a similar way. Their specific AA increased with the 0.3-unit increase in pH, while the specific AA of ascorbic acid and sulphur dioxide in the model solution was similar at both pH values tested. It is known that pH influences polyphenols’ red-ox behaviour, antioxidant capacity and formation of oxidation products [83,84]. Several sources of data are available on the antioxidant capacity of phenol extracts and commercial tannins [4,6,10,11,12,13] but information about their antioxidant capacity at different pH levels is scarce.

The specific antioxidant capacity of SO_2_ in the DPPH assay was the lowest at both pH values. The differences observed in the antiradical activity of the commercial products and extracts are a consequence of the nature of the antioxidant compound, the mechanism of the assay used to quantify the abovementioned activity, reaction media and time of contact [13,82,85]. Phenols are known for their potent antiradical activity [84] while very little information is available on SO_2_ antioxidant activity in the DPPH assay. Our results agree with those obtained by Manzocco et al. [86] who observed that sulphur dioxide affects the oxygen consumption rather than the chain-breaking properties of a model wine system. Other authors have observed that on its own sulphur dioxide in the model solution has a very low antioxidant activity in comparison with other antioxidants [87]. On the contrary, the results obtained by Comuzzo et al. [65] who tested the antioxidant capacity of different oenological products by DPPH assay in a hydroalcoholic tartaric buffer at pH 3.2, found sulphur dioxide and ascorbic acid to have a high antioxidant activity level and observed that effectiveness in radical scavenging seems closely connected to the molar concentration of these tested compounds rather than to their concentration expressed in g/L. In our study, the specific antioxidant capacity of ascorbic acid in the DPPH assay was quite low at both pH values. The antioxidant activity of ascorbic acid was tested in the DPPH assay by Villano et al. [75] together with the pure phenolic compounds commonly present in grapes and wines. These authors reported quite a low antioxidant activity for ascorbic acid, but they highlighted that ascorbic acid has a fast rate of reaction. The same authors ascribed an important role in biological systems where free radicals have very short half-life to antioxidant compounds with this characteristic and, therefore, they concluded that ascorbic acid, as well as kaempferol and procyanidins, characterised by a fast rate of reaction, could be considered highly significant antioxidants. Moreover, the antiradical activity of the red and white wines measured in the DPPH assay seemed to be correlated with the total phenol content rather than the SO_2_ concentration of the wine samples [88]. 

The wines’ resistance against oxidation by adding H_2_O_2_, after 1h at 60 °C, was variable in the function of their composition [64,65]. As expected, red wines were more resistant to oxidation compared to white wines which contain fewer phenolic compounds than red wines. The antioxidant capacity of the different commercial products and UG extracts tested in the accelerated oxidative tests gave different results for red and white wines. In the POM and AOI tests, the wines with added sulphur dioxide showed greater resistance to colour oxidation compared to the majority of the other samples; this result is in agreement with those obtained by other authors who observed an increase in resistance against oxidation in the POM and AOI tests proportional to the increase in the dose of SO_2_ added to the white and red wines [64]. In the white wines, only ascorbic acid protected wine colour better than SO_2_. In the red wines, SO_2_ gave the highest protection against the addition of H_2_O_2_ while ascorbic acid gave the lowest protection, probably due to the formation of an additional amount of H_2_O_2_ as a result of the oxidation of the ascorbic acid, as previously hypothesized by other authors [64]. In the drastic oxidative conditions of the POM and AOI tests, commercial tannins had little effect on the red wines and even less on the white wines. These results are in agreement with those obtained by Celotti et al. [64], who tested the antioxidant capacity of condensed and hydrolysable tannins on different types of red and white wine using the POM and AOI tests. These authors only observed the protective effect of tannins in a sample of young red wines. The amount of UG extracts needed to protect the colour of the wine was much higher than that of the commercial tannins and other products. The results obtained in the POM and AOI tests appear not to be entirely in agreement with what happens in winemaking. Indeed, it is known that commercial tannins have a protective effect against the oxidation of juice and wine [4,6,89] through different mechanisms that include inhibition of polyphenol oxidase (EC 1.14.18.1) and laccase (EC 1.10.32.2), the directly consume of oxygen, radical scavenger activity, metal chelating activity and stabilization of red wine colour [4,5,6,90]. We can assume that the results of the POM and AOI tests obtained with the commercial tannins and UG extracts are due to the drastic experimental oxidative conditions that do not normally occur during winemaking processes. Probably, the accelerated oxidation tests are not actually able to predict the antioxidant capacity of commercial products and UG extract. For these reasons, despite the results of the POM and AOI tests, in the small-scale trials we used the doses of products usually added to wines.

In the small-scale trials, a similar trend of increasing percentages of A420 was observed in the white wine samples with added commercial tannins and Viognier UG extract. After 90 days, the sample with added SO_2_ showed a higher increase in absorbance than the other samples, probably due to the consumption of the SO_2_ added at the start [91]. In the case of the red wine, the addition of SO_2_ was responsible for the decrease in absorbance at 520 nm and CI, due to the bleaching effect of SO_2_ on the free anthocyanins [92]. Then the evolution of A520 was very similar in all of the samples, with a reversible reduction in colour observed 60 days after the second decantation. These results indicated that the addition of commercial tannins and UG extracts could be an interesting alternative to SO_2_ during the ageing and storage of wine. Other authors proposed natural extracts with a high stilbene content obtained from vine residues and highlighted that the use of these products can efficiently contribute to lowering the dose of SO_2_ [10,11]. Similar results were obtained by Esparza et al. [13] who used a stem extract on red wine in the fermentation stage in comparison with SO_2_ and a commercial wood extract. Moreover, it was recently confirmed that oenological tannins exert a protection effect against grape juice and wine oxidation because they have antioxidant activity, they consume directly oxygen, and they exert an inhibitory effect on the laccase activity. Oenological tannins also exert a co-pigmentation effect which can improve and protect the colour of red wines (90).

## 5. Conclusions

This work aimed to characterise natural extracts from unripe grapes and use them for the protection of wine colour during ageing. The results of this study highlighted that the antioxidant activity of natural extracts from UGs can be influenced by small variations in pH in the range of wine pH. This indicates that the dose of the extract should be modulated according to this parameter of wine. The results obtained showed that it is very difficult to predict the actual effect on wine of an antioxidant extract using the accelerated oxidative POM and AOI tests. Recent works suggest that measuring the oxygen consumption rate could be a more suitable way to test the antioxidant capacity of tannins. Promising results were obtained using UG extracts on small volumes of both white and red wine during ageing. However, it is very important to obtain more information about the use of antioxidant extracts from UGs in the other phases of winemaking. The results of this study contribute to the knowledge on the use of natural extracts to lower or even replace SO_2_ in wine ageing. Moreover, the use of natural extracts obtained from wine industry waste could contribute to both wine healthiness and farm sustainability.

## Figures and Tables

**Figure 1 foods-10-01499-f001:**
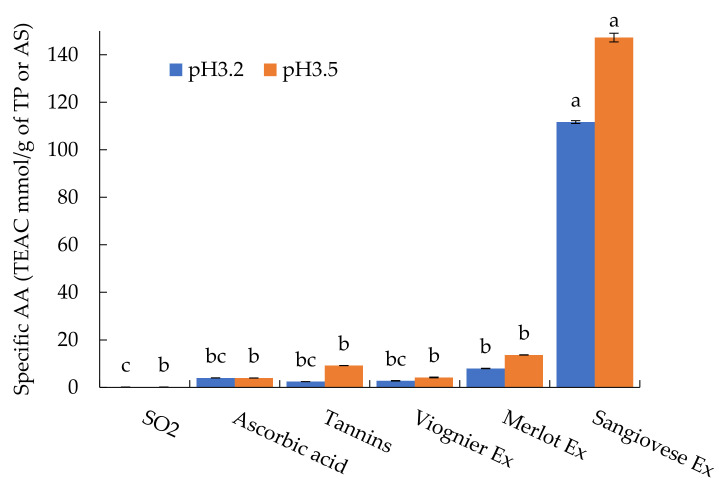
Specific antioxidant activity of the UG extracts (Ex) and commercial products at pH 3.2 and 3.5. Values are the mean ± SD (*n* = 3). Bars represent standard error. Different letters indicate significantly different values among the samples (*p* < 0.001).

**Figure 2 foods-10-01499-f002:**
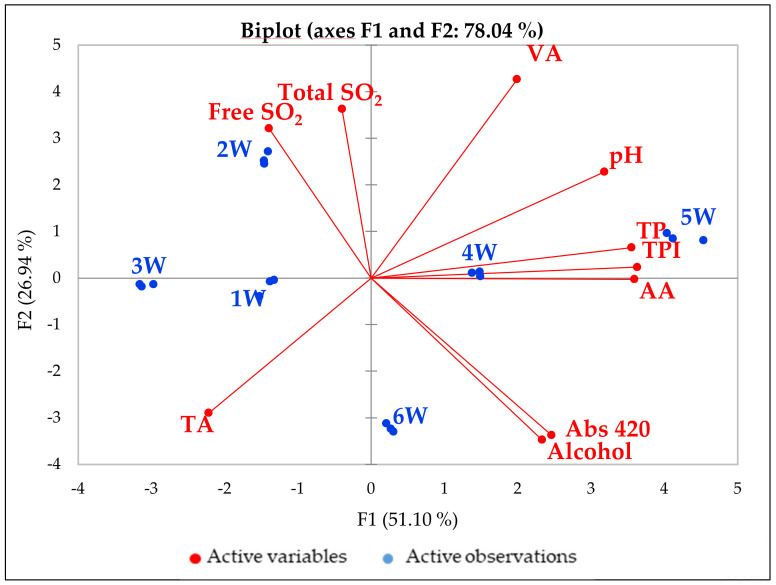
PCA of the 6 white wines. Total acidity (TA); volatile acidity (VA); total phenol (TP); total phenol index (TPI280); antioxidant activity (AA).

**Figure 3 foods-10-01499-f003:**
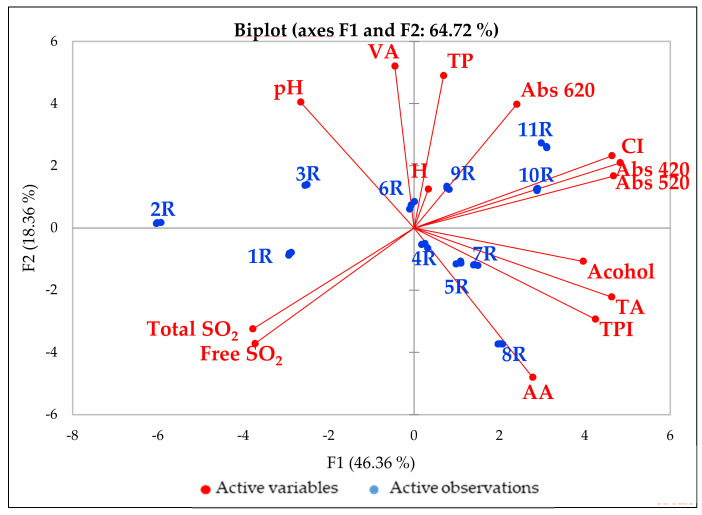
PCA of the 11 red wines. Total acidity (TA); volatile acidity (VA); total phenol (TP); total phenol index (TPI280); antioxidant activity (AA); colour intensity (CI); hue of colour (H).

**Figure 4 foods-10-01499-f004:**
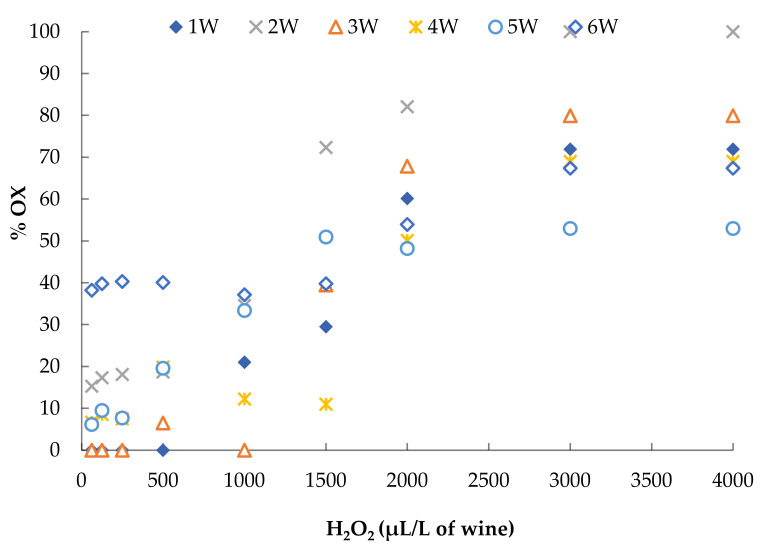
Results of POM test performed with increasing doses of 3% H_2_O_2_, from 62.5 to 4000 μL/L of white wine.

**Figure 5 foods-10-01499-f005:**
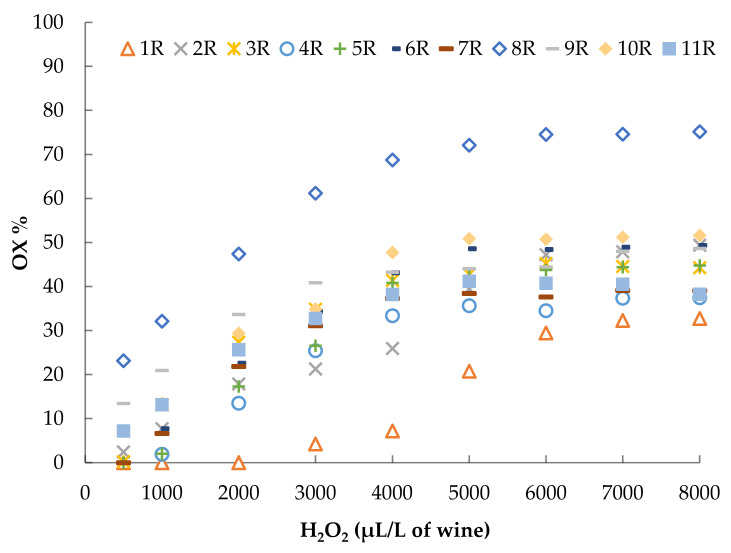
Results of the AOI test performed with increasing doses of 3% H_2_O_2_, from 500 to 8000 μL/L of red wine.

**Figure 6 foods-10-01499-f006:**
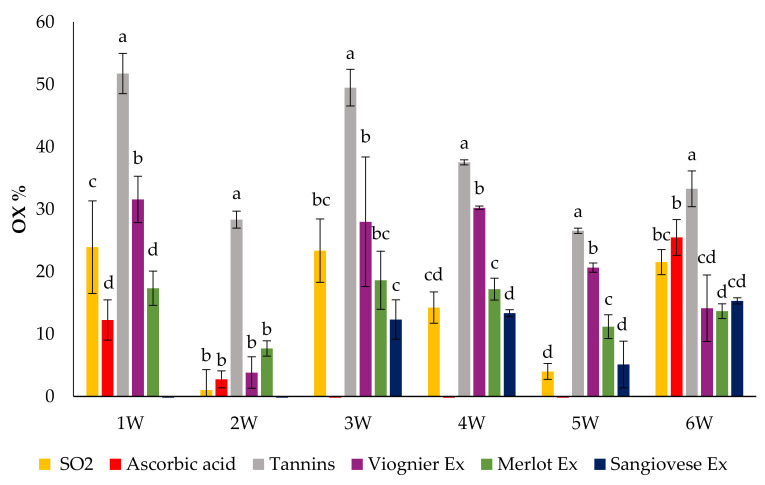
Percentage of colour oxidation of 6 white wines in the POM test performed after the addition of the UG extracts and commercial products. Values are the mean ± SD (*n* = 3). Bars represent standard error. Different letters indicate significantly different values among the samples (*p* < 0.001).

**Figure 7 foods-10-01499-f007:**
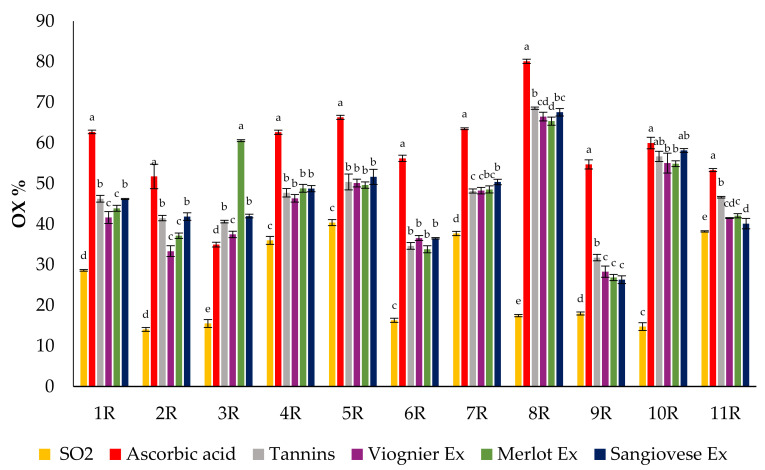
Percentage of colour oxidation of 11 red wines in the AOI test performed after the addition of the UG extracts and commercial products. Values are the mean ± SD (*n* = 3). Bars represent standard error. Different letters indicate significantly different values among the samples (*p* < 0.001).

**Figure 8 foods-10-01499-f008:**
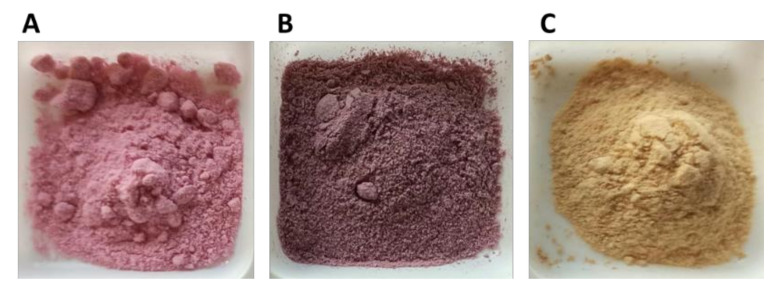
Sangiovese (**A**), Merlot (**B**) and Viognier (**C**) UG extracts.

**Figure 9 foods-10-01499-f009:**
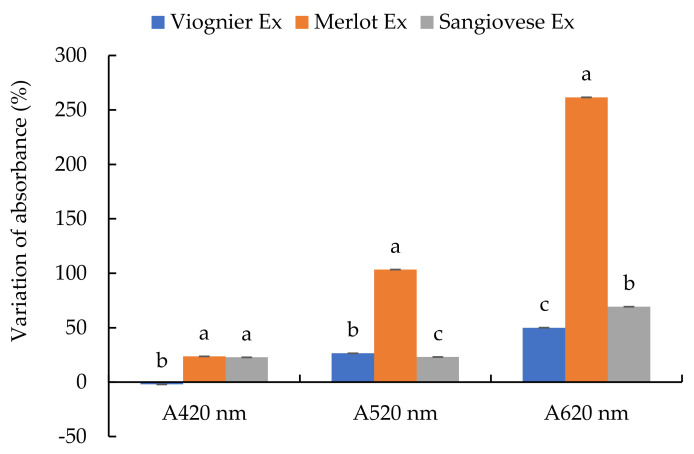
Percentage variation of absorbance at 420, 520 and 620 nm of a white wine after the addition of the UG extracts at a ratio of 400 mg/L. Values are the mean ± SD (*n* = 3). Bars represent standard error. Different letters indicate significantly different values among the samples (*p* < 0.001).

**Figure 10 foods-10-01499-f010:**
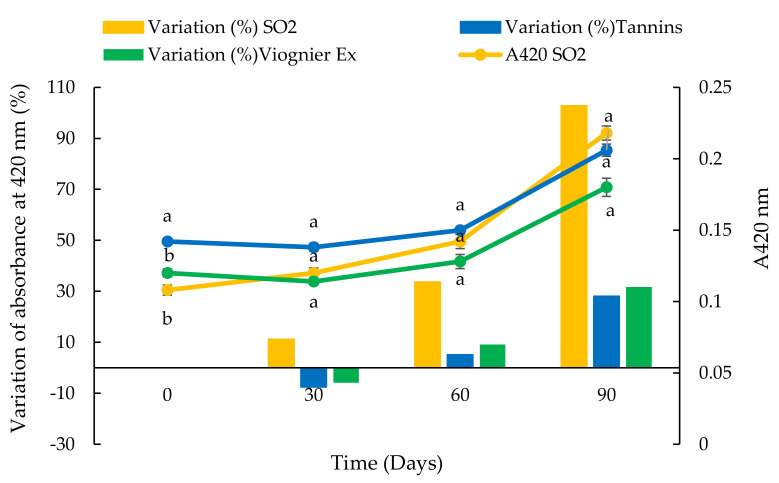
Evolution of absorbance at 420 nm and variation percentage of absorbance at 420 nm of a white wine over 90 days in stainless steel tanks after the addition of SO_2_, commercial tannins and Viognier UG extract. Values are the mean ± SD (*n* = 2). Bars represent standard error. Different letters indicate significantly different values among the samples (*p* < 0.001).

**Figure 11 foods-10-01499-f011:**
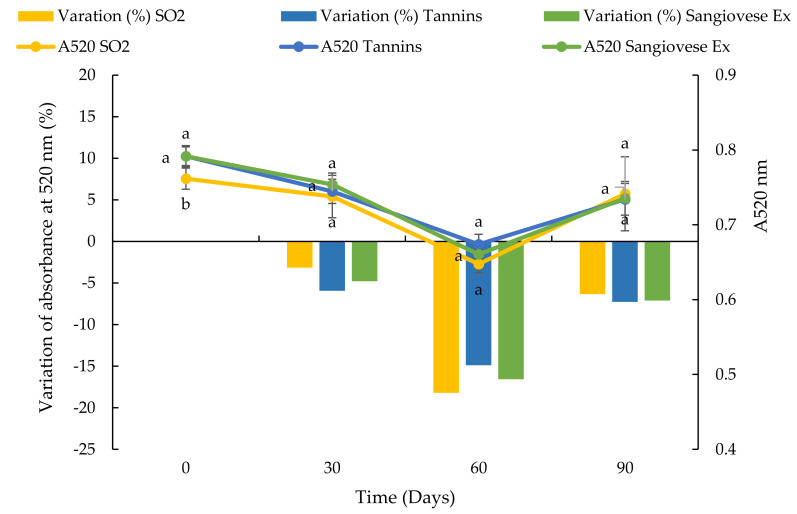
Evolution of absorbance at 520 nm and percentage variation of absorbance at 520 nm of a red wine over 90 days in a stainless-steel tank after the addition of SO_2_, commercial tannins and Sangiovese UG extract. Values are the mean ± SD (*n* = 2). Bars represent standard error. Different letters indicate significantly different values among the samples (*p* < 0.001).

**Figure 12 foods-10-01499-f012:**
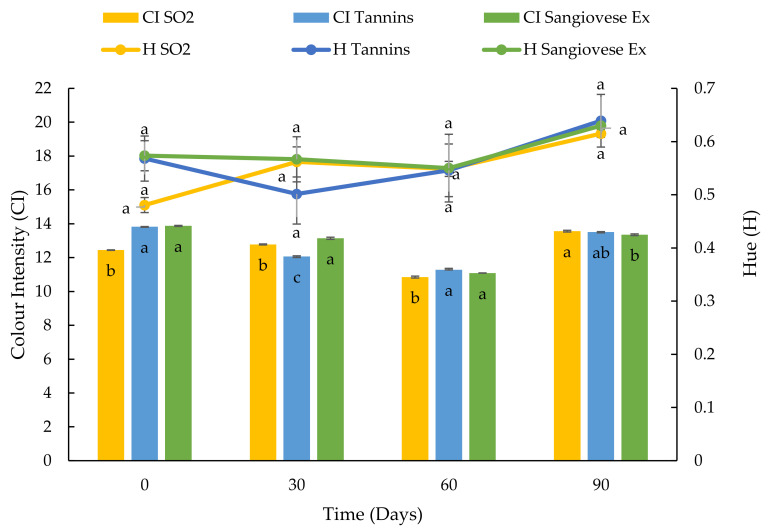
Evolution of colour intensity and hue of a red wine over 90 days in a stainless-steel tank after the addition of SO_2_, commercial tannins and Sangiovese UG extract. Values are the mean ± SD (*n* = 2). Bars represent standard error. Different letters indicate significantly different values among the samples (*p* < 0.001).

**Table 1 foods-10-01499-t001:** Composition of the UG extracts.

Compound	Viognier Ex	Merlot Ex	Sangiovese Ex
Gallic acid	3.0 ± 0.1b	1.4 ± 0.1c	9.5 ± 0.3a
Caffeic acid	1.1 ± 0.1b	4.8 ± 0.6a	0.8 ± 0.0b
Ferulic acid	110.4 ± 28.0a	83.6 ± 11.5a	29.4 ± 4.7b
Coumaric acid	1.1 ± 0.3b	6.7 ± 0.6a	0.5 ± 0.1b
Coutaric acid	147.4 ± 6.2b	99.8 ± 4.3c	191.4 ± 4.6a
Caftaric acid	1190 ± 252a	1061 ± 105.0a	290 ± 73.4b
Fertaric acid	81.0 ± 14.7a	25.4 ± 4.0b	27.6 ± 4.4b
ΣPhenolic acids	1534	1283	549.7
(−)-Epicatechin	9.3 ± 0.8b	1.9 ± 0.2b	64.2 ± 11.9a
(+)-Catechin	91.4 ± 0.8b	20.3 ± 0.6c	327.4 ± 18.9a
Epicatechin-*O*-gallate	1.9 ± 0.1a	0.6 ± 0.0c	1.0 ± 0.0b
ΣFlavan-3-ols	102.6	22.9	392.6
Procyanidin B1	12.9 ± 1.5b	5.3 ± 0.7c	19.0 ± 1.5a
Procyanidin B2	2.5 ± 0.2b	3.0 ± 0.2b	10.1 ± 1.0a
ΣProcyanidins	15.4	8.3	29.1
Quercetin	1.6 ± 0.0a	1.5 ± 0.1a	1.3 ± 0.0b
Isorhamnetin	0.7 ± 0.0a	0.5 ± 0.0b	0.7 ± 0.0a
Quercetin-3-*O*-glucuronide	27.7 ± 1.9b	5.5 ± 0.6c	56.6 ± 3.1a
Quercetin-3-*O*-hexoside	2.6 ± 0.1b	2.0 ± 0.3b	11.8 ± 0.8a
Rutin	0.2 ± 0.0b	0.2 ± 0.0b	0.4 ± 0.0a
Kaempferol	0.6 ± 0.0a	0.3 ± 0.0c	0.5 ± 0.0b
Kaempferol-3-*O*-glucoside	1.1 ± 0.1a	0.3 ± 0.1b	1.1 ± 0.1a
Myricetin	nd	0.7 ± 0.0	nd
Myricetin-*O*-hexoside	1.3 ± 0.1a	1.0 ± 0.1b	1.1 ± 0.1ab
ΣFlavonols	35.9	12.0	73.4
Resveratrol	2.7 ± 0.1b	10.2 ± 1.4a	0.2 ± 0.0c
ΣPhenol compounds	1691	1336	1045
GRP	125.0 ± 12.5a	25.7 ± 2.4b	26.5 ± 1.2b
Glutathione	5.9 ± 0.0a	5.9 ± 0.0a	nd
GSSG	nd	nd	6.6 ± 0.2

Phenolic compounds as μg/g of powder; nd, not detected. Values are the mean ± SD (*n* = 3). Different letters indicate significantly different values within the row (*p* < 0.001).

**Table 2 foods-10-01499-t002:** Percentage relative antioxidant activity (RAA) as radical scavengers of commercial products and UG extracts in the DPPH assay, at two different pH levels, with reference to SO_2_.

pH	SO_2_	Ascorbic Acid	Tannins	Viognier Ex	Merlot Ex	Sangiovese Ex
3.2	100	4725	5907	13123	37602	527559
3.5	100	4493	21518	19074	62239	670644

## Data Availability

Not applicable.

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
