# Peer review of "Characterisation of Extracts Obtained from Unripe Grapes and Evaluation of Their Potential Protective Effects against Oxidation of Wine Colour in Comparison with Different Oenological Products"

_foods, 2021, doi:10.3390/foods10071499_

Round 1
Reviewer 1 Report
The article is interesting and presents interesting results. The antioxidant activity is not so easy to verify and compare because there are many methods doing different approaches.
Why you chose only DPPH method to see antioxidant activity, ABTS and Hydroxyl Radical could also give interesting information’s.
In the section 2.2 the preparation of grapes extracts was different for Sangiovese why!? Why was used a different quantity of Arabic gum, the different preparation could influence the results?
Section 2.5. Why you do not refer Merlot extract.
Concerning the usual analysis, present alcohol content with 1 decimal number, SO2 without decimal number, pH 2 decimal numbers, total acidity 1 decimal number and volatile acidity 2 decimal numbers.
In figure 6 and 7 and more in the work why the statistical treatment is not indicated with letters, to if there are significant differences.
Try do better discussion and compare your results with recent works, there is many works with tannins from Fernando Zamora and Teissedre for example.
The grape extract could eventually be a useful tool for winemaking why you tested in wine inside of musts!? The anti-microbiological activity was not verified as well antioxidasic activity, that’s why is relative abusive say that could be an alternative to SO2.
In the conclusions you said that perhaps is better to see the antioxidant activity the oxygen consumption, why you did not this method?
Try presents more specific conclusions and the principal results should be summarized. Some results should be also indicated in the abstract (numbers, % decreases or increased).
Verify:
Fig. 3. Colour
Put along the work the p value in italic.
Author Response
The article is interesting and presents interesting results. The antioxidant activity is not so easy to verify and compare because there are many methods doing different approaches.
Why you chose only DPPH method to see antioxidant activity, ABTS and Hydroxyl Radical could also give interesting information’s.
Thanks for the comment. The authors agree with the reviewer’s considerations. In fact, it is well known that differences in the level of antioxidant activity can be highlighted when it is assayed with different methods. However, in the literature there are many works that deal with different methods to test antioxidant activity of grapes and their by-products while, to our best knowledge, there are very few works that deal with the capacity of accelerated oxidation tests to predict the effects of different antioxidant products. The authors evaluated the antioxidant activity by DPPH, Folin-Ciocalteau and accelerated oxidation tests, thinking that these three methods were sufficient to screen the UG extracts. Moreover, from the oenological point of view, it was more important to test the effect of extracts in wine by the accelerated oxidation tests than screen them by ABTS, FRAP, ecc.... In fact, nowadays, the accelerated oxidation tests are the most common tools that can be routinely used to predict oxidation in actual winemaking conditions.
In the section 2.2 the preparation of grapes extracts was different for Sangiovese why!? Why was used a different quantity of Arabic gum, the different preparation could influence the results?
Yes, it does influence the results. The Sangiovese extract was dehydrated by spray-drying while Merlot and Viognier by lyophilisation after the sugar removal. The amount of arabic gum depends on the soluble solids, drying technique and type of industrial installation. The authors think that it was interesting to test the effects of different types of extract rather than to compare different varieties of grapes.
Section 2.5. Why you do not refer Merlot extract.
After the characterization of the extracts, the authors decided to choose the Sangiovese extract for the red wine ageing on small-scale because this extract had antioxidant activity at pH 3.5 (the usual pH of red wine) higher than that of the Merlot extract. The better performance of the Sangiovese compared to the Merlot extract was probably due to the richness in flavonoids, as the authors stated in the discussion section.
Concerning the usual analysis, present alcohol content with 1 decimal number, SO2 without decimal number, pH 2 decimal numbers, total acidity 1 decimal number and volatile acidity 2 decimal numbers.
Thanks for the corrections. The authors provided to change decimal numbers in the text.
In figure 6 and 7 and more in the work why the statistical treatment is not indicated with letters, to if there are significant differences.
The authors added in figure 6, 7, 10, 11 and 12 the statistical treatment indicated with letters.
Try do better discussion and compare your results with recent works, there is many works with tannins from Fernando Zamora and Teissedre for example.
The authors have already discussed the results under the light of recent works that deal with exogenous tannins as antioxidants. Some examples are reported below:
Pascual, O.; Vignault, A.; Gombau, J.; Navarro, M.; Gómez-Alonso, S.; García-Romero, E.; Canals, J. M.; Hermosín-Gutíerrez, I.; Teissedre, P. L.; Zamora, F. Oxygen Consumption Rates by Different Oenological Tannins in a Model Wine Solution. Food Chem. 2017, 234, 26–32. https://doi.org/10.1016/j.foodchem.2017.04.148.
Vignault, A.; Pascual, O.; Gombau, J.; Jourdes, M.; Moine, V.; Fermaud, M.; Roudet, J.; Canals, J. M.; Teissedre, P.-L.; Zamora, F. New Insight about the Functionality of Oenological Tannins; Main Results of the Working Group on Oenological Tannins. BIO Web Conf. 2019, 12, 02005. https://doi.org/10.1051/bioconf/20191202005.
Motta, S.; Guaita, M.; Cassino, C.; Bosso, A. Relationship between Polyphenolic Content, Antioxidant Properties and Oxygen Consumption Rate of Different Tannins in a Model Wine Solution. Food Chem. 2020, 313 (July 2019), 126045. https://doi.org/10.1016/j.foodchem.2019.126045.
Raposo, R.; Chinnici, F.; Ruiz-moreno, M. J.; Puertas, B.; Cuevas, F. J.; Carbú, M.; Guerrero, R. F.; Ortíz-somovilla, V.; Moreno-rojas, J. M.; Cantos-villar, E. Sulfur Free Red Wines through the Use of Grapevine Shoots : Impact on the Wine Quality. Food Chem. 2018, 243, 453–460. https://doi.org/10.1016/j.foodchem.2017.09.111.
(12) Raposo, R.; Ruiz-moreno, M. J.; Garde-cerdán, T.; Puertas, B.; Moreno-rojas, J. M.; Gonzalo-diago, A. Effect of Hydroxytyrosol on Quality of Sulfur Dioxide-Free Red Wine. Food Chem. 2016b, 192, 25–33. https://doi.org/10.1016/j.foodchem.2015.06.085.
(13) Esparza, I.; Martínez-Inda, B.; Cimminelli, M. J.; Jimeno-Mendoza, M. C.; Moler, J. A.; Jiménez-Moreno, N.; Ancín-Azpilicueta, C. Reducing SO2 Doses in Red Wines by Using Grape Stem Extracts as Antioxidants. Biomolecules 2020, 10 (10), 1–15. https://doi.org/10.3390/biom10101369.
However, the authors added a sentence in the discussion section (see lines 668-672): “Moreover, it is recently confirmed that oenological tannins really exert a protection effect against grape juice and wine oxidation because they have antioxidant activity, they consume directly oxygen and they exert an inhibitory effect on the laccase activity. Oenological tannins also exert a copigmentation effect which can improve and protect the colour of red wines (90).”
The grape extract could eventually be a useful tool for winemaking why you tested in wine inside of musts!? The anti-microbiological activity was not verified as well antioxidasic activity, that’s why is relative abusive say that could be an alternative to SO2.
The authors plan to test the protective effect also during the other phases of winemaking in the future. However, in this work, the authors tested the UG extracts only during ageing and specified always in the text that the UG extracts could be an interesting substitute of sulphur dioxide during ageing, not in pre-fermentative or fermentative phases. It is true that the authors did not verify the anti-microbial activity of the extracts but they checked the microbial population in the wine during the ageing phase and did not find any product alterations.
In the conclusions you said that perhaps is better to see the antioxidant activity the oxygen consumption, why you did not this method?
Some evidences on the oxygen consumption rate as an index to classify oenological tannins in terms of antioxidant capacity have been collected only in recent years. Emerging techniques, such as oxygen consumption rate or voltammetry, are very promising but require specific instruments (see Motta et al. 2020) and, in the case of the measure of oxygen consumption rate, long times. For these reasons, the authors preferred to check the antioxidant activity with easy and rapid spectrophotometric methods.
Try presents more specific conclusions and the principal results should be summarized. Some results should be also indicated in the abstract (numbers, % decreases or increased).
These are matters of preference. The authors did not want to be repetitive, so they summarised only what they thought necessary for the conclusions. The principal results can be found in the results section. There are many examples of abstracts in the journal that do not report data and numbers. For these reasons, the authors will not change the text.
Verify:
Fig. 3. Colour
Put along the work the p value in italic.
Thanks for the comments. The authors verified and changed the text as suggested.
Reviewer 2 Report
Presented manuscript meets all requirements. Needs only some minor modifications, shown below:
Line 282 I suppose it should be Table 1 instead of Table 2.
Line 284 I think “(Table 1)” could be removed from the text consequently.
Line 317 I would insert “(Figure 1)” in the end of the sentence in Line 318.
Figure 4 I suppose there should be 1W-6W in the figure legend instead of 1B-6B
Figure 12 Following the information contained in the text and figure caption, there should be CI/H Sangiovese Ex instead of CI/ H Merlot Ex in the figure legend.
Author Response
Presented manuscript meets all requirements. Needs only some minor modifications, shown below:
Line 282 I suppose it should be Table 1 instead of Table 2.
Thanks. You are right. The authors changed it.
Line 284 I think “(Table 1)” could be removed from the text consequently.
Thanks. The authors removed it.
Line 317 I would insert “(Figure 1)” in the end of the sentence in Line 318.
Thanks. The authors inserted it.
Figure 4 I suppose there should be 1W-6W in the figure legend instead of 1B-6B
Yes, you are right. The authors corrected the figure legend.
Figure 12 Following the information contained in the text and figure caption, there should be CI/H Sangiovese Ex instead of CI/ H Merlot Ex in the figure legend.
The authors corrected the figure legend.
Reviewer 3 Report
This research examines the potential of using Unripe Grape extracts against the oxidation of wine colour. The hypothesis is tested using routine methods, and basic statistical analysis. The strong point of the paper is the discussion section, as the authors provide very interesting and useful insight on the results and the reasons behind contradictory outcomes produced by different methods.
Line 65 Cluster thinning is not only used for red wines, it can be used for red and whites alike, and it also helps adjust vine balance and achieve a certain crop load and quality level. A reference could be added
Line 150 If the grapes were first crushed and then destemmed, some phenolics from the stems may have been extracted. As this is not discussed in the text the steps should be in their correct order, e.g., first destemmed, and then crushed. In any other case some additional comments should be added regarding phenols from green parts
lines 151-152 “After racking” instead of “after racking off”
Line 172 Was it 500 mg/l or 100 mg/L as it is referred later in the text?
Line 177 which red and white wines are you referring to
Line 181 do you mean that you “used a 300 L volume of both the red and white wines by transferring aliquots (50 L each) into stainless steel tanks. Each trial was arranged in duplicate”? It is not very clear
Author Response
This research examines the potential of using Unripe Grape extracts against the oxidation of wine colour. The hypothesis is tested using routine methods, and basic statistical analysis. The strong point of the paper is the discussion section, as the authors provide very interesting and useful insight on the results and the reasons behind contradictory outcomes produced by different methods.
Line 65 Cluster thinning is not only used for red wines, it can be used for red and whites alike, and it also helps adjust vine balance and achieve a certain crop load and quality level. A reference could be added.
The authors modified the text adding the reference 17 and modified the text as follows: “Unripe grapes (UGs) are a waste product deriving from thinning, which is a green pruning practice carried out on the vines to improve the composition of the grapes for the production of high-quality red and white wine [16-17].”
(17) Keller, M., Mills, L. J., Wample, R. L., Spayd, S. E. (2005) Cluster thinning effects on three deficit-irrigated Vitis vinifera cultivars. American Journal of Enology and Viticulture 56(2), 91–103. https://doi.org/10.1071/CH9540055.
Line 150 If the grapes were first crushed and then destemmed, some phenolics from the stems may have been extracted. As this is not discussed in the text the steps should be in their correct order, e.g., first destemmed, and then crushed. In any other case some additional comments should be added regarding phenols from green parts
The authors corrected the sentence. The grapes are first destemmed and then crushed.
lines 151-152 “After racking” instead of “after racking off”
Corrected.
Line 172 Was it 500 mg/l or 100 mg/L as it is referred later in the text?
The assay in model solution at different pH values was performed at the dose of 500 mg/L. Later in the text 100 mg/L was indicated as the usual oenological dose of some antioxidants.
Line 177 which red and white wines are you referring to
Red wine was Sangiovese and white was Trebbiano. Both the wines were not finished.
Line 181 do you mean that you “used a 300 L volume of both the red and white wines by transferring aliquots (50 L each) into stainless steel tanks. Each trial was arranged in duplicate”? It is not very clear
The authors modified the text as follows: “We used a 300 L volume of both the red and white wines. Each trial was arranged in duplicate, by transferring aliquots (50 L each) into six stainless steel tanks for each wine.”
Reviewer 4 Report
Dear autors
Very interesting publication. It deals with an important aspect of by-product management in viticulture.
Unripe grapes (UGs) are a waste product of vine cultivation that is rich in natural antioxidants. These antioxidants could be used in winemaking as alternatives to SO2. Three extracts were obtained by maceration from Viognier, Merlot and Sangiovese UGs. The composition and antioxidant activity of the UG extracts were studied in model solutions at different pH levels. The capacity of the UG extracts to protect wine colour was evaluated in accelerated oxidation tests and small-scale trials on both red and white wines during ageing in comparison with sulphur dioxide, ascorbic acid and commercial tannins.
The scope of work does not raise any objections. The methods were chosen correctly. The results were statistically processed and presented graphically. The conclusions adequately summarize the obtained results.Author Response
Dear autors
Very interesting publication. It deals with an important aspect of by-product management in viticulture.
Unripe grapes (UGs) are a waste product of vine cultivation that is rich in natural antioxidants. These antioxidants could be used in winemaking as alternatives to SO2. Three extracts were obtained by maceration from Viognier, Merlot and Sangiovese UGs. The composition and antioxidant activity of the UG extracts were studied in model solutions at different pH levels. The capacity of the UG extracts to protect wine colour was evaluated in accelerated oxidation tests and small-scale trials on both red and white wines during ageing in comparison with sulphur dioxide, ascorbic acid and commercial tannins.
The scope of work does not raise any objections. The methods were chosen correctly. The results were statistically processed and presented graphically. The conclusions adequately summarize the obtained results.
Thanks for the comments.